# Application of Condition Monitoring for Hydraulic Oil Using Tuning Fork Sensor: A Study Case on Hydraulic System of Earth Moving Machinery

**DOI:** 10.3390/ma15217657

**Published:** 2022-10-31

**Authors:** Hong-Gyu Jeon, Jong-Kyun Kim, Seon-Jun Na, Min-Seok Kim, Sung-Ho Hong

**Affiliations:** 1Test & Assessment Technology Team, Hyundai Construction Equipment, Yongin 16891, Korea; 2Digital Technology Team, Hyundai Construction Equipment, Seongnam 13591, Korea; 3Department of Mechanical System Engineering, School of Creative Convergence Engineering, Dongguk University-WISE Campus, Gyeongju 38066, Korea

**Keywords:** hydraulic oil, oil sensor, earthmoving machinery, oil analysis, condition monitoring

## Abstract

In this study, we focus on the correctness of oil condition monitoring, specifically of a tuning forks sensor in hydraulic systems. We also aim to analyze the correlation between the online monitoring sensor signal and offline oil analysis by periodically sampling the hydraulic oil. In recent years, condition-based monitoring (CBM) of hydraulic oils has played a key role in extending earthmoving machinery uptime and reducing maintenance costs. We performed rig test and field test to develop a condition monitoring system based on oil analysis for construction equipment. Using the rig test, a reference line for the diagnosis of viscosity and dielectric constant for the new hydraulic oil was derived, and the characteristics of each sensor parameter for artificial contamination and oxidation were confirmed. In order to affirm the validity of oil diagnosis using oil sensors, the oil sensors were applied to four excavators to detect changes in oil conditions over 12 months. It was found that monitoring hydraulic oil with an oil sensor detecting the change in oil properties and contamination can provide reliable information for establishing diagnostic criteria. The finding allows us to predict the remaining oil life and to determine the oil change intervals based on the diagnosis of the oil condition.

## 1. Introduction

The hydraulic system is the fundamental working system of earthmoving machinery and has the primary function of transmitting energy and lubricating the friction parts. A monitoring system of hydraulic oil condition is essential for earthmoving machinery to achieve greater efficiency, reduce maintenance costs, and prevent unexpected system downtime. Hydraulic oil plays an important role in lubricating, sealing, preventing wear and corrosion, as well as cooling. It is a combination of base oils and additives consisting of complex mixtures of hydrocarbons [1]. The quality of the hydraulic oils also deteriorates with time of being used in earthmoving machinery. The main reasons for hydraulic oil degradation are contamination, thermal degradation, and oxidation. Some contaminants come from impurities of hoses, pipes, and hydraulic parts or wear debris from the lubricated contact surfaces of the rotating parts, as well as water and dust from the outside. Water contamination, in particular, promotes the hydrolysis of additives and speeds up the oxidation of hydraulic oil [2]. Oxidation is the predominant reaction of hydraulic oils in a running machine that creates a major problem with oil performance. Oxidation causes problems such as an increase in viscosity and acid number, formation of varnish and sludge, depletion of additives, loss of resistance to foaming, and demulsification along with corrosion [3]. Oxidized oils lead to performance problems in hydraulic systems by expanding the electronic valve to improve efficiency and control accuracy. However, these valves have a very low level of tolerance to oil contamination and, in particular, varnish. Varnish is a submicron soft contaminant that can form sludge or surface deposits. The sticky film resulting from varnish can further damage the sliding surfaces, attracting other hard contaminants such as dust and wear particles, which contribute to problems with valve spools, journal bearings, plugging filters, increased friction during parts movement, and reduced cooling system performance.

Currently, hydraulic oil condition diagnostics can be divided into online monitoring and oil sampling analysis, known as oil services. The oil service is carried out by periodic oil sampling and appropriate laboratory test methods. The oil service mainly covers kinematic viscosity, atomic emission spectroscopy (AES), acid number (AN), particle counter, water content, and membrane patch colorimetric (MPC). Kinematic viscosity is a measure of the fluid’s internal resistance to flow under the influence of gravitational forces and is an indicator that determines the lubricant film thickness [4]. Viscosity lowered below the criteria can lead to an increased failure rate caused by wear due to metal-to-metal contact wear. AES can monitor the type and content of wear debris, contaminants, and additives in hydraulic oil [5]. The particle counting test indicates the cleanliness of the hydraulic oil and is related to the filtration performance of the return filter. The oxidation of hydraulic oil can be determined by the acid number and MPC tests. The AN test is one of the oil analysis methods used to assess the degree of additive depletion, acid contamination, and oxidation. AN does not directly determine the rate of oxidation but only measures the oxidation by-product. It is also advantageous to trend AN in order to determine the depletion rate of certain additives. Several methods are available to determine the water contamination level in hydraulic oils. One of the most accurate methods is the Karl Fischer coulometric titrator (KF). Unlike other techniques, KF can trace low levels of water contamination and prevent side reactions with additives in hydraulic oils using an evaporator [6]. MPC varnish potential testing is an essential analytical test to determine the propensity of a lubricant to form varnish deposits. MPC patch testing consists of diluting an oil sample with a strong solvent and vacuum filtering the mixture through an ultra-white, fine-pored membrane. The color of the membrane patch is an indicator of the varnish potential range. The more yellow and black the patch, the greater the vanishing potential. Oil sampling analysis provides more detailed information about the conditions of hydraulic oils and earthmoving machinery’s operation. However, it requires on-site visits for periodic oil sampling and is likely to miss inter cycle events due to its long detection cycle. Therefore, the lack of real-time monitoring of hydraulic oils affects the maintenance strategy of earthmoving machinery.

The main purpose of maintenance is to preserve the functionality of the system in an economical manner [7,8]. In order to achieve this goal, the functionality of equipment should be delivered consistently with high-level performance and reliability. In addition, maintenance should reach the designed output level and have a flow chart with minimal resource costs and within limitations of system condition and safety. Condition-based maintenance plays an important role in the diagnosis and prognosis of machine deterioration. The decision about the maintenance is based on an evaluation of the system state by analyzing the real-time or periodic information about that condition [9,10].

Proper monitoring of the condition of the hydraulic systems can ensure high efficiency, reducing maintenance costs and preventing unexpected failures [11]. Lubricant condition monitoring is one of the condition-based maintenance techniques known as oil analysis and plays a key role in the management of aging and deteriorating machinery. Since the monitoring of lubricant condition is initiated based on the state of the machinery, the components are only replaced when the deterioration rate exceeds the criteria. As a result, unplanned machine downtime can be minimized. Furthermore, the ability to perform predictive maintenance means that the total cost of ownership can be reduced as the service life of the hydraulic components and machines can be fully utilized. Therefore, original equipment manufacturers (OEMs) or maintenance service providers can also plan their service schedules more accurately by knowing the correct maintenance information [12].

This study presents an approach to the validation of monitoring the hydraulic oil condition in earthmoving machinery using an oil sensor based on the correlation between periodic oil analysis and the sensor output signal. We focused on the actual acquisition of monitoring data typically used in mobile hydraulic systems (i.e., earthmoving machinery). The relevance of the actual field test for verification of the sensor signal can be proved by numerous facts. Firstly, not only the drastic change in hydraulic oil flow rate and temperature but also external vibrations caused by various operation conditions affect the variation of the sensor signal. Therefore, it is necessary to establish a data analysis procedure that takes into account the characteristics. Secondly, it is crucial to verify whether the sensor represents the condition of hydraulic oil over a long period of time by comparing the sensing data with the oil analysis results. It relies on an approximately 2000-h oil condition monitoring study on a 14-ton hydraulic wheeled excavator to identify the method to improve a hydraulic system maintenance strategy, namely through effective oil condition and contamination monitoring.

## 2. Sensors Available for Hydraulic Oil Monitoring

Recently, many studies have been carried out on developing oil monitoring sensors that can provide continuous and real-time monitoring of equipment health. Several types of sensors can be applied to monitor the oil quality. Commercially available oil sensors can measure oil properties, including wear debris, water content, viscosity, aeration, soot, particle count, etc., based on capacitive, inductive, acoustic, and optical methods. Table 1 introduces typical oil sensors for monitoring the condition of hydraulic oils.

### 2.1. Oil Viscosity Sensors

Viscosity is the next critical parameter of the condition of the lubricant and the mechanical system. It represents the performance of the lubricating oil, which is to ensure a sufficient thickness of the lubricating film between the two contact surfaces. The inappropriate viscosity of lubricating oils leads to excessive wear and frictional heat, unnecessary fatigue, adhesion, and even catastrophic failure [30,31]. A widely used method for viscosity sensors is the detection of body displacement [13], acoustic [14,15,16], and vibration detection [17,18]. The performance and reliability of displacement sensors depend on the presence of moving pistons and environmental conditions. Another way of measuring viscosity is based on acoustic waves. These viscometers use piezoelectric materials to generate acoustic waves. Although the acoustic wave sensor has a simple structure and thermal stability [32], the electrode is unresistant to the acid component of the lubricant, which reduces its accuracy and service life. Heinisch et al. [18] developed a viscosity sensor that uses commercially available tuning forks to generate magnetic force and, thus, mechanical excitation. The recorded and evaluated frequency responses of tuning forks can be used to measure the viscosity. The method has the advantages of low cost, robustness, and ease of installation and use. However, it is sensitive to vibrations and electrical noise from the outside.

### 2.2. Oil Contamination Sensors

Statistics [33] show that from 70% to 80% of failures in hydraulic systems can be attributed to contaminated oil. Dust, wear debris, and water are among many other types of contamination that can lead to hydraulic system failures, water being probably the most common [34]. It occurs in hydraulic oils in three different forms: dissolved, emulsified, and free [19]. Water in an emulsified or free form not only causes excessive wear debris by reducing the load-bearing capacity of the lubricating film but also rapidly oxidizes hydraulic oils and hydrolyzes additives. The water content sensor measures the relative humidity or the oil saturation level. The capacitive method is a widely used industrial method for measuring water content. Its main advantages are low cost, wide measuring range, and simplicity. However, this method can be easily affected by the environment, e.g., temperature, other contamination, and relative humidity, which can be correlated with the oil saturation level [20]. The particle counter can determine the number of small particles in a hydraulic system and represents the levels of cleanliness of the hydraulic oil in accordance with ISO 4406. The ISO code number scale is used for each particular size to represent amounts within a specified range of particles. Lubrication problems are caused, in particular, by contamination of particles smaller than 40 μm. As these particles have a similar size to the lubricating film between the two contact surfaces, overpressure can easily occur in the elastohydrodynamic lubrication regime [35].

The particle counting method for online monitoring of contaminants in the oil mainly includes resistive, inductive, acoustic, capacitive, and optical methods. Each methodology has its strengths and weaknesses, such as the evaluation of particle types, the differentiation between ferrous and non-ferrous metals, and the sensitivity of detection [22,23,24,25,27,28,36,37].

### 2.3. Wear Debris Sensors

Detecting small wear debris is critical to diagnosing abnormal lubrication conditions and predicting the wear or failure of hydraulic components. The debris affects the wear mechanism, such as accelerated abrasive wear, and indicates a certain condition of the mechanical system in the lubricating component. The wear debris in hydraulic oil remains in a constant concentration and small size under normal working conditions. When the machine operates in abnormal conditions, the concentration and size of wear particles increase and differ from normal [38,39]. Therefore, the detection of wear debris in the oil is an important and effective method of fault diagnosis, essential for the scheduled maintenance of key components in the hydraulic system. Depending on the detection method, the monitoring of wear debris present when changing the lubricating oil can be divided into inductive, capacitive, acoustic, and optical methods. However, the above methods have their merits and demerits. The inductance method can clear up ferrous particles, but the detection sensitivity of non-ferrous particles is low. Particle size and the distinction between air bubbles and solid particles are the focus of the acoustic method. The disadvantage of this method, however, is difficulty in distinguishing a metallic solid from a dielectric solid because all solid particles have similar acoustic reflectance coefficients [40]. The sensitivity of the optical method is high [26], but it is affected by the transparency of the oil, and the configuration of the monitoring system is complicated. Capacitive methods are widely used owing to their good temperature stability, simple structure, great adaptability, and high throughput. However, this method has limitations such as low sensitivity, water influence, and low throughout. Although this method has disadvantages, it is still the most effective and practical method for numerous applications [41].

In the construction equipment industry, machinery is often exposed to harsh operating conditions and inadequate daily maintenance, which can lead to accelerated oil degradation and hydraulic system wear. The degradation of hydraulic oil, including water and dust contamination, is a very complex and complicated process. Assessing the oil state by monitoring a single property, such as viscosity, water content, particle counting, wear debris concentration, etc., is not a sufficient method to evaluate its condition. For a comprehensive assessment of hydraulic oil conditions, a multi-quality sensor is used in the hydraulic system of earthmoving machines, i.e., an oil quality sensor or an integrated oil properties sensor [29]. The oil quality sensor directly and simultaneously measures the dynamic viscosity, density, dielectric constant, and temperature, as shown in Figure 1. The specification of this sensor is presented in Table 2. Depending on the tuning fork technology, the sensor monitors the direct and real-time relationship between multiple properties to estimate oil degradation and contamination [42]. The sensor monitoring system enables service providers to schedule necessary maintenance as soon as an abnormal incident is detected. The following part of this paper presents the validity of monitoring hydraulic oil along with the oil analysis by periodic oil sampling, which was carried out for 12 months.

## 3. Experimental Details

### 3.1. Test Rig and Oil Analysis Instrument

In order to establish the diagnostic criteria for oil monitoring, the physical properties of the new hydraulic oil were obtained using a test rig with an oil quality sensor and oil analysis instruments. The test rig consists of a beaker with the tested oil, a hot plate stirrer, a power supply, a CAN-USB module, an oil quality sensor with a harness, and signals data acquisition system, as shown in Figure 2. The monitored oil is ISO VG46 hydraulic oil, and its properties are given in Table 3. As mentioned above, absolute viscosity, density, dielectric constant, and temperature are the physical properties that can be obtained with an oil sensor. The temperature and flow of the hydraulic oil were controlled by a heater and a magnetic stirrer. Additionally, the magnetic bar was rotated at a speed of 100 rpm to generate a corresponding flow. An analysis of the main reasons for hydraulic oil’s failure in earthmoving machinery shows that the state of the oil is abnormal due to oil degradation caused by water contamination and oxidation. In this study, experiments were conducted in order to assess the mixing characteristics of distilled water and the level of varnish resulting from processes such as oxidation. We carried out a total of 4 experiments by using two sensors, two times each. This is to confirm the individual sensor precision and repeatability. The validity of the sensor output can be verified by comparison with the oil analysis results. Methods of hydraulic oil analysis are kinematic viscosity, acid number, water content, atomic emission spectroscopy, and membrane patch colorimetry. International standards (ASTM; American Society for Testing and Material) and test instrument specifications for each oil analysis method are summarized in Table 4. Kinematic viscosity, which is absolute viscosity divided by density, can be directly compared with the result of the oil analysis. On the other hand, the dielectric constant, which is affected by the water contamination and the oxidation of hydraulic oil, can be indirectly compared by an oil analysis method such as water content, acid number, and membrane patch colorimetry.

### 3.2. Monitoring with an Oil Sensor

Proper oil sampling is critical for a successful oil diagnosis. In general, the oil sampling area is in front of the filters and behind hydraulic system components such as the main pump, slewing and travel motor, piston, and valves. The described design ensures an excessive data income about wear debris, depletion of additives, declining viscosity, etc. Additionally, the mounting position of the oil sensor enables taking into account the permissible pressure of the sensor and the continuous oil flow and minimizing the vibration effects caused by earthmoving machinery. A schematic diagram of the excavator’s hydraulic system is shown in Figure 3. The oil sensor was installed downstream from the main pump drain line and upstream from the drain filter in accordance with the sampling guideline. The drain line is used to transfer hydraulic oil used for lubrication and leaked oil during the compression process of cylinder, valve plate, shoe, etc., which are the main components of the main pump. Four excavators with a service life of more than 2000 h after factory shipment were selected, and hydraulic oil monitoring was planned during field endurance testing within 12 months. These excavators have undergone routine service schedules with the return filter replaced every 1000 h of operation, and the hydraulic oil replaced every 5000 h of operation. For this reason, it is effective to select excavators close to the oil change schedule in order to monitor the diagnostic condition and degradation characteristics of the hydraulic oil. Periodic oil sampling was performed at the time of initial sensor installation and every 500 operating hours (See Table 5).

## 4. Results and Discussion

### 4.1. New Oil Analysis from the Test Rig and Oil Analysis Instrument

Figure 4 shows the kinematic viscosity at varying temperatures using the ASTM D445 viscometer test method. The correlation between temperature and kinematic viscosity was estimated by carrying out measurements for 20 samples taking into account the viscosity dispersion of the new oil. The regression analysis (Equation (1)) was conducted using the Vogel model, which is one of the most accurate and useful in engineering calculations [43]. The upper and lower prediction limits represent the range of dispersion of the experimental data. In the formula, *η* is kinematic viscosity, and *T* is the temperature in Kalvin degrees. The accuracy of the oil sensor signal for viscosity was confirmed by comparing the viscosity vs. temperature measurement with the results of the regression analysis from the viscometer values of the new oil. These signals were obtained with two sensors due to the variability in accuracy and are shown in Figure 5. In contrast to the viscosity verification process, there is no measuring method with an oil analysis instrument for the dielectric constant, so the regression analysis was performed based on the sensor signal for the new hydraulic oil and determined as a diagnostic criterion. The regression model for the dielectric constant is a simplified Debye equation [44], as shown in Figure 6 and Equation (2). In the formula, *ε* is the dielectric constant, and *T* is the temperature in Kalvin degrees.
(1)$$\eta =0.13285\times e^{\frac{812.108}{\left(T\pm 173.865\right)}}$$
(2)$$\varepsilon =1.54928\times \frac{208.539}{T}$$

### 4.2. Water Contamination and Varnish Test

Figure 7 presents the kinematic viscosity and dielectric constant at different temperatures when mixing distilled water with new oil from 0.5 mL to 1.0 mL. These experiments were carried out once each because repeated tests are impossible due to the evaporation of moisture by heating. A total of three feasibility tests were carried out in advance. After injecting water into the oil, each water content result was obtained using a paint shaker and a Karl Fischer titrator based on ASTM D6304. Varnish contamination measurements of three different hydraulic oils are shown in Figure 8. Three oxidized hydraulic oils were collected from excavators operated for 5591, 6541, and 7521 h beyond the oil change cycle. These oils were subjected to the MPC test to evaluate the degree of oxidation; the results of the oil analysis are summarized in Table 6. It can be seen that the dielectric constant is more effective than the kinematic viscosity in detecting water and varnish contaminations in hydraulic oil. The change in dielectric constant due to water and varnish contamination showed the same characteristics, which increased completely compared to the new oil. The slope of the dielectric constant, depending on the changing temperature, remains the same for varnish-contaminated oil as for new oil. Conversely, for water contamination, a transition point is observed. Based on this difference, a change in the slope of water contamination and varnish contamination can be distinguished in the transition point.

### 4.3. Monitoring with an Oil Sensor

#### 4.3.1. Sensor Data Analysis Method

Hydraulic oil monitoring field tests were conducted for approximately 12 months on four excavators. Figure 9 presents the raw sensor signal data for kinematic viscosity and dielectric constant measured over one working day. In contrast to the rig test, it can be seen that the dispersion is relatively increased due to the sudden change in the flow rate depending on the working conditions and external vibrations. As mentioned in Section 3.1, kinematic viscosity and dielectric constant regression models are the reference line of oil diagnosis. Using this model as a reference line, Figure 10 shows the kinematic viscosity and dielectric constant error levels as a function of each temperature for the excavator sensor signal. The signal data processing method was developed to calculate representative values of the kinematic viscosity and dielectric constant through standard deviation of the error rate and noise filtering, as shown in Figure 11.

#### 4.3.2. Monitoring Data Analysis

Figure 12 shows the results of kinematic viscosity monitoring for four excavators over a 12-month period. From the moment the sensor was installed, it can be seen that the kinematic viscosity of all four excavators has gradually decreased over time. In general, the monitoring criteria for oil condition diagnosis are different for each original equipment manufacturer (OEM) that provide maintenance or service package to customers. In the general industry, the kinematic viscosity of hydraulic oil allows for a viscosity change of about ±15~20% based on new oil. In particular, there are times when the viscosity monitoring signal increases sharply on the Monitoring C excavator, which means a temporary increase in viscosity by refilling new hydraulic oil during repair work on this excavator due to leakage. The accuracy of the kinematic viscosity monitoring was confirmed by comparison with the results of oil analysis from a total of four oil sampling, including the sensor installation. The analysis results of the oil samples are summarized in Table 7. The monitored kinematic viscosity value, which is a representative value of the analyzed signal data at the time of taking the oil samples, was compared with the value measured by the oil analysis and is shown in Table 8. Comparing the sensor and the measured viscosity value, the maximum error rate is 6.1%, and when Monitoring A is excluded, the error rate is less than 5%.

Figure 13 shows the monitoring results of the dielectric constant signal and shows the gradual increase in the dielectric constant for all monitoring excavators. The MPC test results, as shown in Figure 14, also present an increase in varnish contamination and oil darkening. Monitoring A shows a drastic increase in the dielectric constant due to the inflow of deteriorated brake oil during repair work on the axle components. The brake oil of the wheel excavator’s axle is hydraulic oil, which remains stationary for a long time without circulation. The mixing of the severely degraded oils resulted in a peak of the dielectric constant followed by an increase in the overall monitored value of the dielectric constant. The MPC test results for the monitoring excavator showed the same outcomes. All monitoring excavators showed normal conditions for water contamination which affects the dielectric constant, and the acid number indicated oxidation of the hydraulic oil. The results of the oil pollution analysis of both the water and the acid number, which influenced the dielectric constant, met the criteria for hydraulic oil management. In addition, the amount of additives consumption and wear debris was also observed in the normal state. The generation of varnish, which is one of the indicators of chemical degradation of hydraulic oil, cause a gradual increase in the dielectric constant over a long period of time while water contamination and acid number are within the normal state range. It can be seen that the MPC test results gradually increase as all monitoring excavators are operated for a long time. It is possible that dielectric constant monitoring will be used as a method to determine the timing of oil exchange along with viscosity monitoring.

## 5. Conclusions

This paper presents the monitoring of hydraulic oil for earthmoving machinery based on an oil sensor. The criteria for diagnosing the oil condition were established on the basis of the rig test; the monitoring was performed for 12 months with the use of four excavators. From the rig test, a reference line was determined for the kinematic viscosity and dielectric constant, which are the criteria for diagnosing the oil conditions. The effectiveness of the sensor signal was confirmed by comparison with the results of the oil analysis. By monitoring the oil in a wheel excavator, the sensor was able to successfully detect a change in viscosity due to the refilling of the oil and a change in dielectric constant due to the mixing of highly oxidized oil during the component repair process. In addition, the validity of the sensor signal was confirmed by comparing the monitoring results of these sensors with the results of the oil analysis based on samples.

The main empirical reasons for causing abnormal conditions of hydraulic oil in earthmoving machinery are an increase in wear particles due to a decrease in the thickness of lubricating film according to decline viscosity. In addition, there is chemical degradation of hydraulic oil due to long-term operation under high temperatures. The above two oil deteriorations are characterized by very gradual progression. On the other hand, water contamination not only accelerates the generation of wear particles by rapidly decreasing lubrication capacity but also causes the machine to go down. It is possible to detect changes in viscosity and to diagnose the state of oil due to varnish generation by oil monitoring based on the oil sensor. Unexpected machine down can be prevented by detecting sudden changes in the oil state, such as water contamination, which may occur between oil sampling cycles. By detecting the change in viscosity and the degree of varnish generation, the change cycle of the oil may be established based on state diagnosis rather than a fixed change cycle. In light of the evidence obtained, improving the uptime of construction equipment through real-time monitoring, as well as upgrading the maintenance strategy from regular oil change cycles to oil change cycles based on condition diagnostics, is proven worthwhile.

## Figures and Tables

**Figure 1 materials-15-07657-f001:**
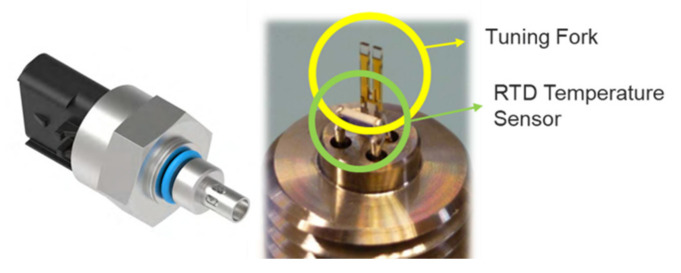
Overall view of tuning fork sensor for hydraulic oil.

**Figure 2 materials-15-07657-f002:**
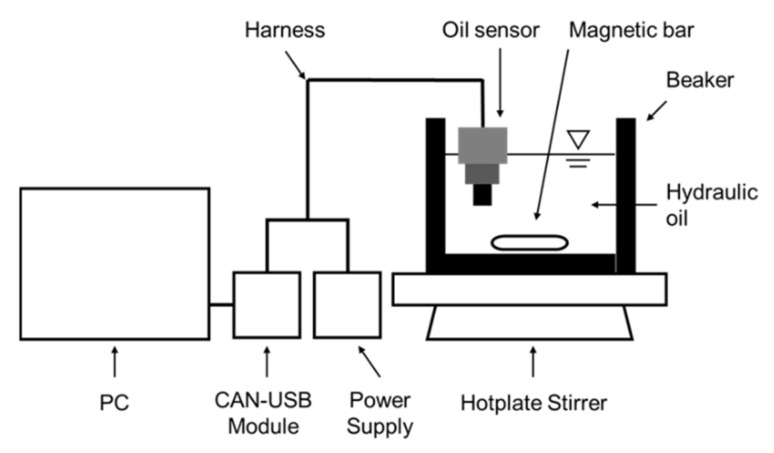
Schematic image of the test rig using by oil quality sensor.

**Figure 3 materials-15-07657-f003:**
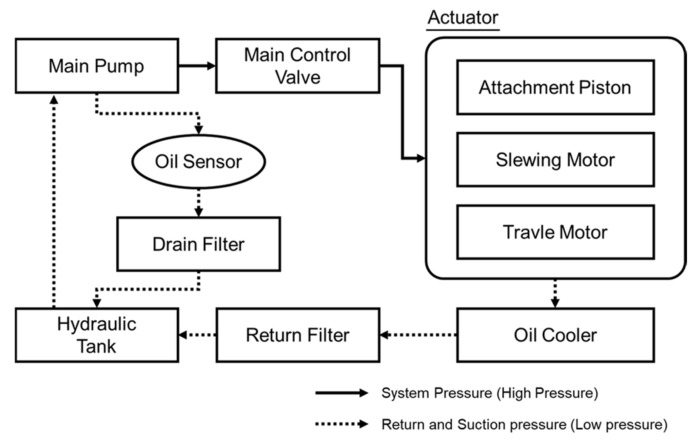
Schematic diagram of the excavator hydraulic system.

**Figure 4 materials-15-07657-f004:**
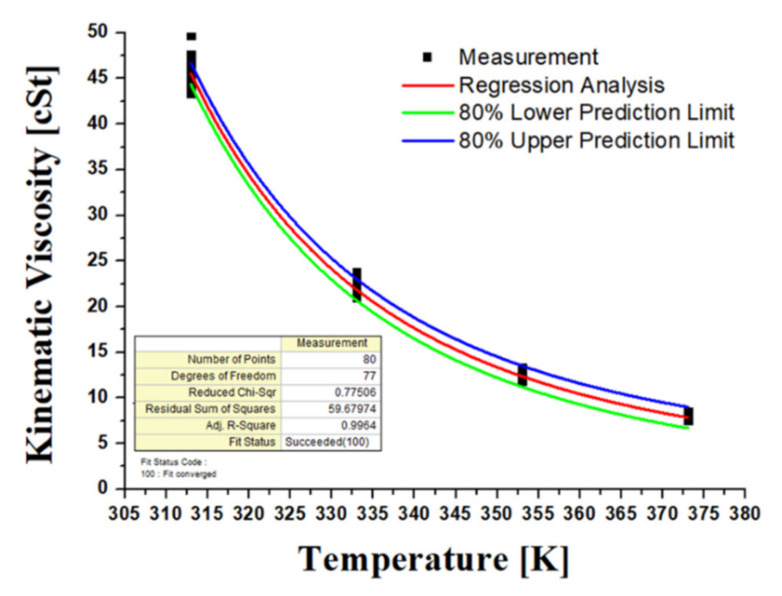
The correlation between kinematic viscosity and temperature using the Vogel model.

**Figure 5 materials-15-07657-f005:**
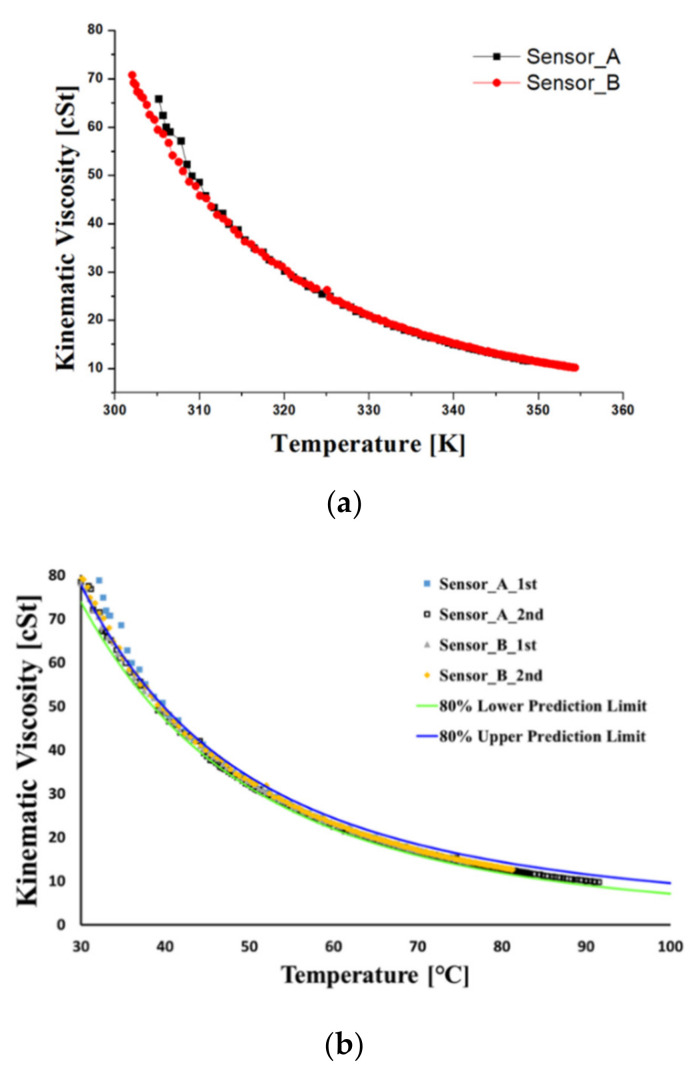
The correlation between kinematic viscosity and temperature using the Vogel model. (**a**) Oil sensor signal of kinematic viscosity at varying temperatures. (**b**) The analysis of regression and prediction limits for kinematic viscosity.

**Figure 6 materials-15-07657-f006:**
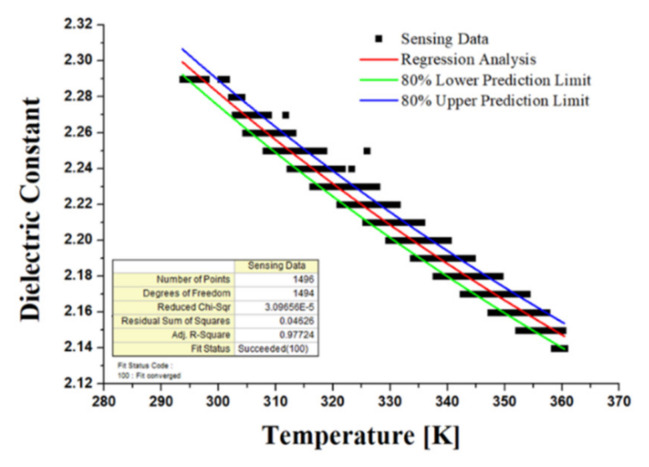
The correlation between dielectric constant and temperature using the Debye model.

**Figure 7 materials-15-07657-f007:**
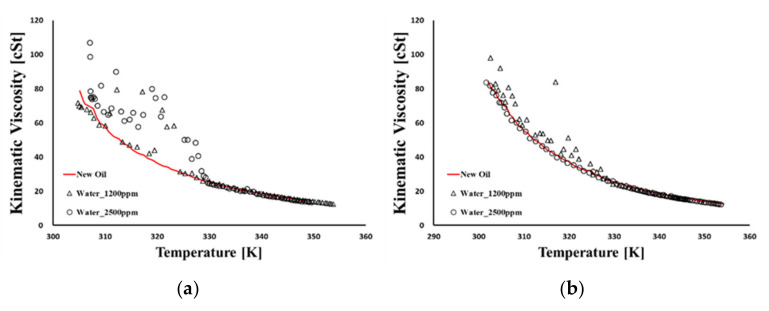
The oil sensor signal of water contamination. (**a**) Sensor A of kinematic viscosity. (**b**) Sensor B of kinematic viscosity. (**c**) Sensor A of dielectric constant. (**d**) Sensor B of dielectric constant.

**Figure 8 materials-15-07657-f008:**
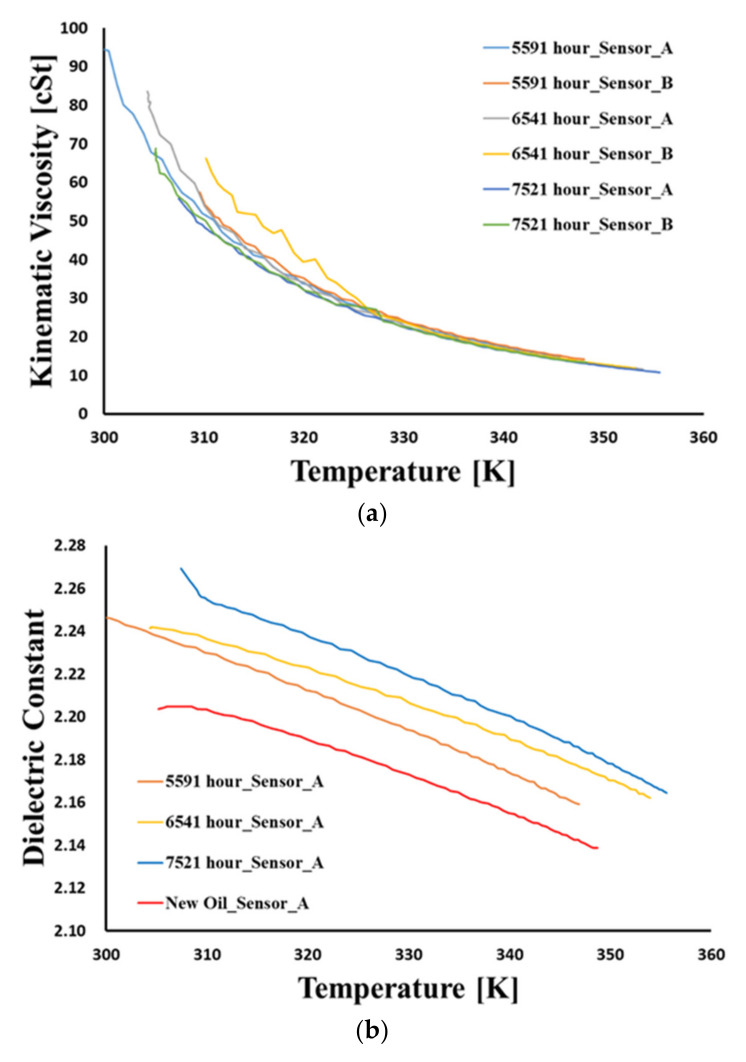
The oil sensor signals of varnish contamination. (**a**) Sensor signals of kinematic viscosity. (**b**) Sensor A signals of dielectric constant. (**c**) Sensor B signals of dielectric constant.

**Figure 9 materials-15-07657-f009:**
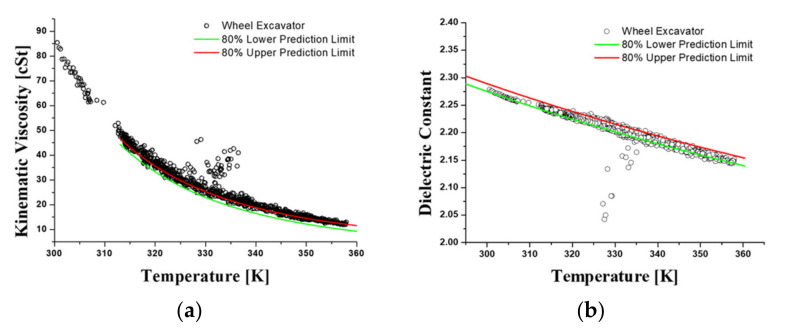
The raw excavator data during one operating day. (**a**) Kinematic viscosity. (**b**) Dielectric constant.

**Figure 10 materials-15-07657-f010:**
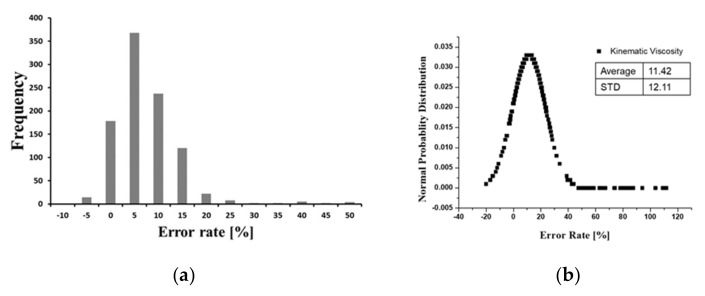
Analysis of error rate variability for excavator’s sensing data during one operating day. (**a**) Error rate frequency of viscosity. (**b**) Normal probability distribution of viscosity. (**c**) Error rate frequency of dielectric constant. (**d**) Normal probability distribution of dielectric constant.

**Figure 11 materials-15-07657-f011:**
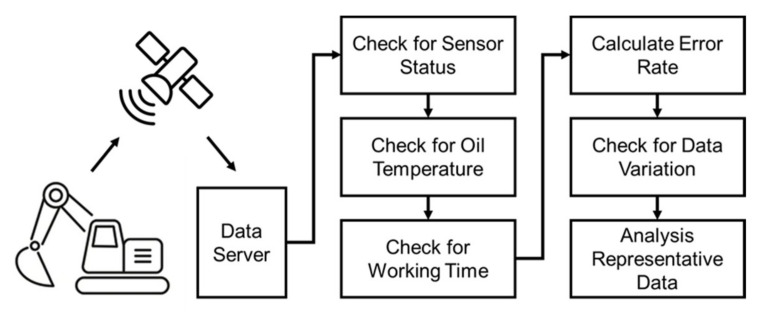
Oil sensor signal data analysis procedure.

**Figure 12 materials-15-07657-f012:**
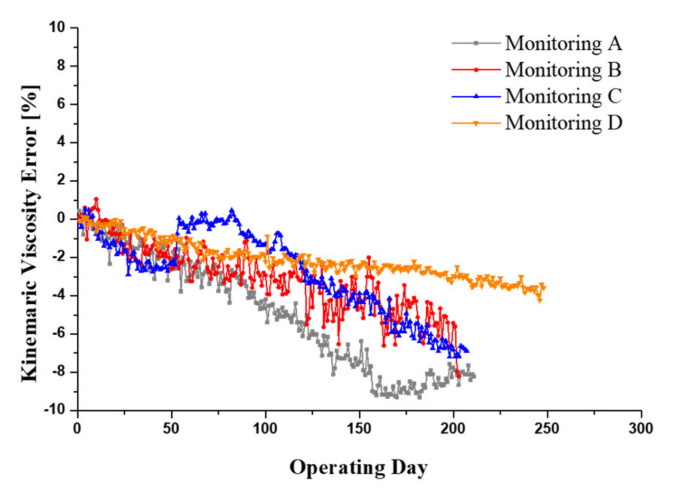
Kinematic viscosity monitoring results of four excavators over a 12 month period.

**Figure 13 materials-15-07657-f013:**
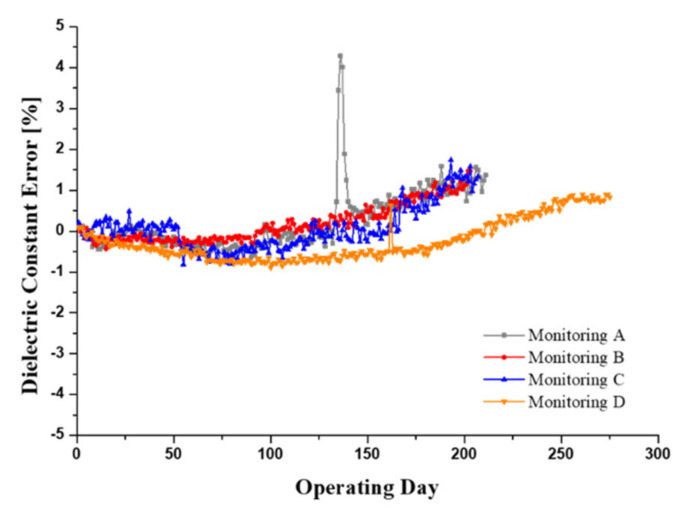
Dielectric constant monitoring results of four excavators over a 12 month period.

**Figure 14 materials-15-07657-f014:**
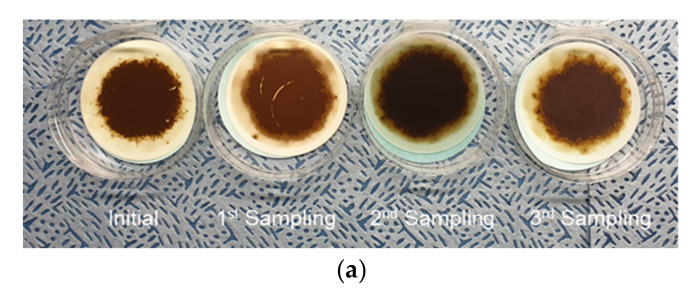
Sample of the MPC varnish potential test. (**a**) Monitoring A. (**b**) Monitoring B. (**c**) Monitoring C. (**d**) Monitoring D.

**Table 1 materials-15-07657-t001:** Oil sensors for hydraulic oil.

Sensor Type	Detect Method	Output Signal	Advantage	Disadvantage	Ref.
Viscosity sensors	Displacement	Dynamic viscosity	Simple structure	Low reliability	[13]
	Acoustic wave		Simple structure, thermal stability	Weak corrosion	[14,15,16]
	Vibration		Low cost and robust	Bubbles influence	[17,18]
Water/ Moisture sensors	Capacitance	Water saturation % Relative humidity %	Low cost and simplicity	Other contamination influence	[19]
	Resistance		High throughput	Other contamination influence	[20]
	Fiber-optical		High sensitivity	Complicated structure	[21]
Wear debris sensors	Inductance	Wear debris size and concentration	Differentiate ferrous and non-ferrous	Low sensitivity	[22]
	Capacitance		High throughput and sensitivity	Water influence	[23]
	Ultrasonic		Detect solid debris and air bubbles	Complicated structure	[24,25]
	Optical		Detect debris size and shape	Affected by oil transparency	[26]
Particle counter sensors	Inductance	Wear debris size and concentration	Individual wear debris	cannot detect non-ferrous	[27]
	Capacitance		High sensitivity	Water influence	[23]
	Ultrasonic		Discriminating air bubbles	cannot differentiate metallic and non-metallic	[24,25]
	Optical		Detect particles shape	Affected by oil transparency	[28]
Integrated sensors	Inductance	Oil cleanliness Viscosity Water Wear debris	Integrated monitoring of multi-properties	Complicated structure and inaccurate due to the compound influence	[29]
	Capacitance				
	Resistance				
	Optical				

**Table 2 materials-15-07657-t002:** Specification of tuning fork sensor (@Temperature = 373 [K]).

Measurement Parameters	Measurement Ranges	Accuracy
Absolute viscosity [cP]	0.5~50	±2%
Density [g/cm^3^]	0.65~1.50	±1%
Dielectric constant	1.0~6.0	±1%
Temperature [K]	233~423	0.1 [K]

**Table 3 materials-15-07657-t003:** Properties of the new ISO VG 46 hydraulic oil.

Kinematic Viscosity [cSt]		Viscosity Index	Acid Number [mgKOH/g]	Water Content [ppm]	Density [g/cm^3^ @15 °C]
@40 [°C]	@100 [°C]				
48.28	8.23	145	0.702	50.855	0.8457
Wear [ppm]		Contamination [ppm]	Additives [ppm]		
Fe	Cu	Si	Ca	P	Zn
0.0	0.0	0.0	74.5	463.2	725.8

**Table 4 materials-15-07657-t004:** Test method and instrument for oil analysis.

Index	Test Method	Model	Remark
Kinematic viscosity	ASTM D445	U-Visc 120	0.5–10,000 [cSt]
Acid number	ASTM D664	T5 Titrator	
Atomic emission spectroscopy	ASTM D6595	Spectroil 120C	24 elements 0–1000 [ppm]
Water content	ASTM D6304	C30SX Titrator with evaporator	0–10,000 [ppm]
Membrane patch colorimetry	ASTM D7843	RM200	

**Table 5 materials-15-07657-t005:** Schedules of periodic oil sampling (Engine operating hours).

	Installation (September 2021)	1st Oil Sampling (January 2022)	2nd Oil Sampling (May 2022)	3rd Oil Sampling (August 2022)
Monitoring A	4495 h	4967 h	5591 h	5848 h
Monitoring B	4697 h	5191 h	5808 h	6380 h
Monitoring C	5450 h	6040 h	6700 h	7163 h
Monitoring D	2109 h	2974 h	3858 h	4641 h

**Table 6 materials-15-07657-t006:** The oil analysis results of water and varnish contamination for the rig test.

Rig Test Oil		Kinematic Viscosity [cSt]	Acid Number [mgKOH/g]	Water Content [ppm]	MPC
New hydraulic oil		47.39	0.713	45.253	-
Water contamination	Distilled water 0.5 mL	47.58	0.695	1284	-
	Distilled water 1.0 mL	47.79	0.724	2494	-
Varnish contamination	5591 h	46.45	0.852	63.260	87.3
	6541 h	46.31	0.887	70.585	89.1
	7521 h	44.91	0.726	64.755	96.4

**Table 7 materials-15-07657-t007:** Results of hydraulic oil analysis based on periodic oil sampling from excavators.

Oil Monitoring		Kinematic Viscosity @40 °C [cSt]	Acid Number [mgKOH/g]	Water Content [ppm]	Wear (Fe/Cu) [ppm]	MPC
New hydraulic oil		48.28	0.702	50.855	0.0/0.0	-
Monitoring A	Install	46.57	0.675	69.255	9.0/4.1	76.4
	1st sampling	46.52	0.927	51.355	9.6/3.7	73.4
	2nd sampling	46.45	0.852	63.260	11.2/3.9	87.3
	3rd sampling	46.31	0.914	60.523	10.5/4.1	77.0
Monitoring B	Install	45.55	0.801	68.580	3.9/4.3	67.7
	1st sampling	45.46	0.872	39.660	4.4/6.1	68.6
	2nd sampling	45.18	0.915	46.651	3.0/5.1	64.9
	3rd sampling	44.84	0.924	55.251	4.2/5.5	71.1
Monitoring C	Install	46.30	0.844	60.025	4.9/1.8	76.5
	1st sampling	46.09	0.939	49.380	5.1/1.6	64.1
	2nd sampling	45.84	0.944	57.541	4.5/1.8	77.3
	3rd sampling	45.75	0.954	62.452	5.2/1.9	75.6
Monitoring D	Install	46.47	0.824	71.620	4.8/5.4	58.6
	1st sampling	46.44	1.030	51.495	6.6/4.9	36.3
	2nd sampling	46.25	1.075	76.950	7.5/5.5	64.6
	3rd sampling	46.28	1.112	72.562	8.4/6.2	90.1

**Table 8 materials-15-07657-t008:** Comparison of sensing and measured value of the kinematic viscosity for excavators.

		Monitoring A				Monitoring B			
		Initial	1st	2nd	3rd	Initial	1st	2nd	3rd
Kinematic viscosity @40 °C [cSt]	Oil sensor	45.57	46.52	46.45	46.31	45.55	45.46	45.18	44.84
	Oil analysis	46.74	45.49	44.14	43.48	46.09	44.93	44.37	43.17
	Error [%]	0.36	−2.23	−4.98	−6.11	1.20	−1.17	−1.79	−3.72
		Monitoring C				Monitoring D			
		Initial	1st	2nd	3rd	Initial	1st	2nd	3rd
Kinematic viscosity @40 °C [cSt]	Oil sensor	46.30	46.09	45.84	45.75	46.47	46.44	46.25	46.28
	Oil analysis	46.32	45.99	45.19	43.51	47.41	46.52	46.22	45.62
	Error [%]	0.05	−0.22	−1.41	−4.89	2.02	0.17	−0.06	−1.42

## Data Availability

Not applicable.

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
