# Peer review of "Application of Condition Monitoring for Hydraulic Oil Using Tuning Fork Sensor: A Study Case on Hydraulic System of Earth Moving Machinery"

_materials, 2022, doi:10.3390/ma15217657_

Round 1
Reviewer 1 Report
the topic is interesting and the authors' arguments are well-posed,
However, some points are unclear, in particular, the experimental setup and the measurement procedures must be made explicit:
- where are the sensors located?
- how many measurements have been carried out for the results of fig. 5 and 7?
- how are the monitoring conditions met?
- what are the metrological characteristics of the chosen sensors?
- imposing maximum and minimum limits on the mean regression curves of the experimental data is not metrologically correct;
On the other hand, measurement uncertainties should be properly assessed in order to estimate the overall accuracy of measurement procedures
measuring.
- English must be revised
Author Response
Response to Reviewer 1
Dear Reviewer
We appreciate your comments and prepared a sincere response to your comments.
Moreover, we received professional English proofreading.
In addition to correcting English expression, we have completed the overall correction of grammar, articles, prepositions and so on.
The revised parts are marked in red in the manuscript.
Sincerely yours,
Sung-Ho Hong
Point 1: where are the sensors located?
Response 1.
We have additional explanation into line 247 to 249.
“The drain line is to transfer hydraulic oil used for lubrication and leaked oil during the compression process of cylinder, valve plate, shoe, etc. which are the main components of the main pump.”
Point 2: how many measurements have been carried out for the results of fig. 5 and 7
Response 2.
In the case of Figure 5, the change in kinematic viscosity according to temperature was measured. We carried out a total of 4 experiments by using two sensors, two times each. This is to confirm the individual sensor precision and repeatability. Oil sensors were installed in four selected excavators for research purpose about oil monitoring after performing lab tests.
In the case of Figure 7, the experiments related to water contamination was carried out once each. This is because in the case of water contamination experiment, repeated tests are impossible due to evaporation of water due to heating. The decrease in water content after the experiment was measured to be about 10 ~ 15 % compared to initial value. A total of three feasibility tests were carried out to identify the characteristics of the dielectric constant according to water contamination in advance.
We added some explanation into line 224 to 226 and line 290 to 292 as follows:
“We carried out a total of 4 experiments by using two sensors, two times each. This is to confirm the individual sensor precision and repeatability.”
“These experiments were carried out once each because repeated tests are impossible due to evaporation of moisture by heating. A total of three feasibility tests were carried out in advance.”
Point 3: how are the monitoring conditions met?
Response 3.
We have additional explanation for viscosity criteria to line 331 to 335 as follows:
“In general, the monitoring criteria for oil condition diagnosis are different for each original equipment manufacturers (OEMs) that provide maintenance or service package to customers. In the general industry, the kinematic viscosity of hydraulic oil allows for a viscosity change of about ± 15 ~ 20 % based on new oil.”
However, water contamination is very rare in the hydraulic system of excavator according to the empirical maintenance of the excavator. In this study, we confirmed the change in the dielectric constant due to the generation of varnish, which is one of the results of oil oxidation. Future work is to evaluate the lifetime of hydraulic oil through analysis of correlation between the dielectric constant and the varnish.
We have added additional explanation to line 359 to 365 as follows:
“The generation of varnish which is one of the indicators of chemical degradation of hydraulic oil, that cause a gradual increase in the dielectric constant over long period time while water contamination and Acid number are within the normal state range. It can be seen that the MPC test results gradually increase as all monitoring excavators are operated for a long time. It is possible that dielectric constant monitoring will be used as a method to determine the timing of oil exchange along with viscosity monitoring.”
Point 4: what are the metrological characteristics of the chosen sensors?
Response 4.
We appreciate your comment. We added Table 2 as follows.
Table 2. Specification of tuning fork sensor (@Temperature = 373 [K])
|
Measurement Parameters |
Measurement Ranges |
Accuracy |
|
Absolute Viscosity [cP] |
0.5 ~ 50 |
± 2 % |
|
Density [g/cm3] |
0.65 ~ 1.50 |
± 1 % |
|
Dielectric Constant |
1.0 ~ 6.0 |
± 1 % |
|
Temperature [K] |
233 ~ 423 |
0.1 [K] |
Point 5: imposing maximum and minimum limits on the mean regression curves of the experimental data is not metrologically correct;
Response 5.
The criteria for the diagnosis of the excavator’s hydraulic oil needed reference line according to temperature. As Reviewer point out, the reference line was derived through regression analysis based on the experimental data which are obtained by oil analysis instrument. But, the maximum and minimum of the prediction limits derived by regression analysis are the result of considering the distribution of experimental data, not the maximum and minimum criterion of the diagnosis criteria of oil. We first confirmed the validity of the sensor on a bench test by whether the sensor signals are distributed within the maximum and minimum range of the aforementioned prediction limit. In addition, by applying a sensor to the excavator and monitoring it, the effectiveness was further confirmed through comparison between the sensor measurements and the oil analysis results.
To clarify, we have additional explanation into line 271 to 272 as follows.
“The upper and lower prediction limits represent the range of dispersion of the experimental data.”
We thank the Reviewers again for their time and insights.
Reviewer 2 Report
The authors tried to do the condition monitoring of hydraulic oil using tuning fork sensor considering case study on hydraulic system of earth moving machine. The paper is interesting and suitable for publication with minor revisions. Please do the following minor corrections
1. Please improve the abstract
2. In table 1, please give details of the list of instruments, range accuracy and percentage uncertainties. (Very important)
3. Please double check the English throughout the manuscript and make the necessary changes
4. Please offer more discussion of the result and how your investigation is giving a new idea in this research are through comparison to other research papers.
5. Between Line 190 to Line 205, please see if you can refer to any literature.
6. Fig.8 (c) C should be small letter
Author Response
Response to Reviewer 2
Dear Reviewer
We appreciate your comments and prepared a sincere response to your comments.
Moreover, we received professional English proofreading.
In addition to correcting English expression, we have completed the overall correction of grammar, articles, prepositions and so on.
The revised parts are marked in red in the manuscript.
Sincerely yours,
Sung-Ho Hong
Point 1: Please improve the abstract
Response 1.
We appreciate this comment. We modified the abstract as follows.
In this study, we focus on the correctness of oil condition monitoring in hydraulic systems; in this case the use of a tuning forks sensor. We also aimed to analyze the correlation between the online monitoring sensor signal and off-line oil analysis by periodically sampling the hydraulic oil. In recent years, condition-based monitoring (CBM) of hydraulic oils has played a key role in extending earthmoving machinery uptime and reducing maintenance costs. We performed rig test and field test to develop condition monitoring system based on oil analysis for construction equipment. Using the rig test, a reference line for diagnosis of viscosity and dielectric constant for the new hydraulic oil has been derived, and the characteristics of each sensor parameter for artificial contamination and oxidation have been confirmed. In order to affirm the validity of oil diagnosis using oil sensors, the oil sensors have been applied to four excavators to detect changes in oil conditions during 12 months. It has been found that monitoring the hydraulic oil with the oil sensor, in other words, detecting the change in oil properties and contamination, can provide reliable information for establishing diagnostic criteria. This enables to the prediction of the remaining oil life and determination of oil change intervals based on the diagnosis of the oil condition.
Point 2: In table 1, please give details of the list of instruments, range accuracy and percentage uncertainties. (Very important)
Response 2.
Unfortunately, we could not find any information about the detailed specifications of the sensors. In Table 1, we added advantages and disadvantages according to the type of sensor. Moreover, we added related papers as references as follows.
|
Type of sensor |
Detect method |
Output signal |
Advantage |
Disadvantage |
Ref |
|
|
Viscosity sensors |
Displacement |
Dynamic viscosity |
Simple structure |
Low reliability |
[15] |
|
|
Acoustic wave |
Simple structure, thermal stability |
Weak corrosion |
[16-18] |
|||
|
Vibration |
Low cost and robust |
Bubbles influence |
[19, 20] |
|||
|
Water/ Moisture sensors |
Capacitance |
Water saturation %, Relative humidity % |
Low cost and simplicity |
Other contamination influence |
[24] |
|
|
Resistance |
High throughput |
Other contamination influence |
[25] |
|||
|
Fiber-optical |
High sensitivity |
Complicated structure |
[27] |
|||
|
Wear debris sensors |
Inductance |
Wear debris size and concentration |
Differentiate ferrous and non-ferrous |
Low sensitivity |
[31] |
|
|
Capacitance |
High throughput and sensitivity |
Water influence |
[30] |
|||
|
Ultrasonic |
Detect solid debris and air bubbles |
Complicated structure |
[32, 33] |
|||
|
Optical |
Detect debris size and shape |
Affected by oil transparency |
[39] |
|||
|
Particle counter sensors |
Inductance |
Wear debris size and concentration |
Individual wear debris |
cannot detect non-ferrous |
[28] |
|
|
Capacitance |
High sensitivity |
Water influence |
[30] |
|||
|
Ultrasonic |
Discriminating air bubbles |
cannot differentiate metallic and non-metallic |
[32, 33] |
|||
|
Optical |
Detect particles shape |
Affected by oil transparency |
[35] |
|||
|
Integrated sensors |
Inductance |
Oil cleanliness Viscosity, Water, Wear debris |
Integrated monitoring of multi-properties |
Complicated structure and inaccurate due to the compound influence |
[41] |
|
|
Capacitance |
|
|||||
|
Resistance |
|
|||||
|
Optical |
|
|||||
Point 3: Please double check the English throughout the manuscript and make the necessary changes
Response 3.
We corrected grammatical errors and checked English.
Point 4: Please offer more discussion of the result and how your investigation is giving a new idea in this research are through comparison to other research papers.
Response 4.
The section 4.3.2 and Conclusion have been rewritten to emphasize the originality of this paper as follows.
“The generation of varnish which is one of the indicators of chemical degradation of hydraulic oil, that cause a gradual increase in the dielectric constant over long period time while water contamination and Acid number are within the normal state range. It can be seen that the MPC test results gradually increase as all monitoring excavators are operated for a long time. It is possible that dielectric constant monitoring will be used as a method to determine the timing of oil exchange along with viscosity monitoring.”
“The main empirical reasons for causing abnormal conditions of hydraulic oil in earthmoving machinery are an increase in wear particles due to a decrease in the thickness of lubricating film according to decline viscosity. In addition, there is chemical degradation of hydraulic oil due to long-term operation under high temperature. The above two oil deteriorations are characterized by very gradual progression. On the other hand, water contamination not only accelerates the generation of wear particles by rapidly decreases lubrication capacity, but also causes machine down. It is possible to detect changes in viscosity and to diagnose the state of oil due to varnish generation by oil monitoring based on the oil sensor. Unexpected machine down can be prevented by detecting sudden changes in oil state such as water contamination which may occur between oil sampling cycles. By detecting the change in viscosity and the degree of varnish generation, the exchange cycle of the oil may be established based on state diagnosis rather than a fixed exchange cycle.”
Point 5: Between Line 190 to Line 205, please see if you can refer to any literature
Response 5.
We have added the references as follows.
In the construction equipment industry, machinery is often exposed to harsh working conditions and inadequate daily maintenance, which can lead to accelerated oil degradation and hydraulic wear. The degradation of hydraulic oil, including water and dust contamination, is a very complex and complicated process and it is inadequate to evaluate its condition just monitoring a single property, such as viscosity, water content, particle counting, and wear debris concentration, etc. For a comprehensive assessment of hydraulic oil condition, a multi-quality sensor is used in the hydraulic system of earthmoving machines, i.e. an oil quality sensor or an integrated oil properties sensor [41]. The oil quality sensor directly and simultaneously measures the dynamic viscosity, density, dielectric constant and temperature, as shown in Figure 1. The specification of this sensor are lied in Table 2. Depending on the tuning fork technology, the sensor monitors the direct and real-time relationship between multiple properties to estimate degradation and contamination of the oils [42]. The sensor monitoring system enables service providers to schedule necessary maintenance as soon as an abnormal incident is detected. The following part of this paper presents the validity of hydraulic oil monitoring along with the oil analysis by periodic oil sampling, which was carried out for 12 months of monitoring the oil condition and degradation.
Point 6: Fig.8 (c) C should be small letter
Response 6.
We corrected error as follows.
(c) Sensor B signals of dielectric constant
We thank the Reviewers again for their time and insights.

Round 2
Reviewer 1 Report
The authors have satisfied all the questions opened in my review. It is my opinion that now the article is suitable to be published.